# Replicating and extending the effects of auditory religious cues on dishonest behavior

**Aaron D. Nichols**[1,2☯]*, **Martin Lang**[3☯], **Christopher Kavanagh**[4,5‡], **Radek Kundt**[3‡], **Junko Yamada**[5‡], **Dan Ariely**[2‡], **Panagiotis Mitkidis**[2,6‡]

**1** Questrom School of Business, Boston University, Boston, MA, United States of America, **2** Social Sciences Research Institute, Duke University, Durham, NC, United States of America, **3** LEVYNA Laboratory for the Experimental Research of Religion, Masaryk University, Brno, Czech Republic, **4** Institute of Cognitive & Evolutionary Anthropology, University of Oxford, Oxford, United Kingdom, **5** Department of Behavioral Science, Hokkaido University, Sapporo, Japan, **6** Department of Management, Aarhus University, Aarhus, Denmark

☯ These authors contributed equally to this work.
‡ These authors also contributed equally to this work.
* aarondavidnichols@gmail.com

**Data Availability Statement:** All relevant data are within the paper and its Supporting Information files.

**Funding:** ML and RK acknowledge funding by LEVYNA Laboratory for the Experimental Research

## Abstract

Although scientists agree that replications are critical to the debate on the validity of religious priming research, religious priming replications are scarce. This paper attempts to replicate and extend previously observed effects of religious priming on ethical behavior. We test the effect of religious instrumental music on individuals' ethical behavior with university participants (N = 408) in the Czech Republic, Japan, and the US. Participants were randomly assigned to listen to one of three musical tracks (religious, secular, or white noise) or to no music (control) for the duration of a decision-making game. Participants were asked to indicate which side of a vertically-bisected computer screen contained more dots and, in every trial, indicating that the right side of the screen had more dots earned participants the most money (irrespective of the number of dots). Therefore, participants were able to report dishonestly to earn more money. In agreement with previous research, we did not observe any main effects of condition. However, we were unable to replicate a moderating effect of self-reported religiosity on the effects of religious music on ethical behavior. Nevertheless, further analyses revealed moderating effects for ritual participation and declared religious affiliation congruent with the musical prime. That is, participants affiliated with a religious organization and taking part in rituals cheated significantly less than their peers when listening to religious music. We also observed significant differences in cheating behavior across samples. On average, US participants cheated the most and Czech participants cheated the least. We conclude that normative conduct is, in part, learned through active membership in religious communities and our findings provide further support for religious music as a subtle, moral cue.

of Religion [CZ.1.07/2.3.00/20.048] and the Czech Science Foundation (GA CR) [18-18316S]. The studies were funded by the Center for Advanced Hindsight at Duke University https://advanced-hindsight.com/. Yes, one of the authors, Dr. Dan Ariely, is the principal investigator at the Center for Advanced Hindsight. Dr. Ariely advised on the writing of the manuscript, but did not participate in study design, data collection, or decision to publish.

**Competing interests:** NO. The authors have declared that no competing interests exist.

## Introduction

Religious systems use various emotionally charged symbols to induce individual normative behavior. For example, in Judaism, the sound of the *shofar* (a ram's horn) is a spiritual alarm, indicating it is time for people to repent and ask God for forgiveness. In the Muslim tradition, the call to prayer, *athan*, is a regular reminder to pray and re-establish the norms of communal life that Allah stipulated through his prophet Muhammad. Importantly, previous studies have suggested that such subliminal religious reminders induce prosocial and normative behavior. However, the research documenting the effects of religious priming on prosocial behavior has recently faced substantial criticism [1–3]. To contribute additional empirical data to this ongoing debate, we investigate the effects of religious auditory cues on individual honesty and extend the research literature on religious priming by adapting the methodology employed by Lang and colleagues [4]. Using a more generalizable sample and a less biased cheating task, we replicate important conceptual results observed by Lang and colleagues [4] yet fail to find evidence for the anticipated moderating effect of self-reported religiosity. Our results add important nuance to previous results by investigating how the efficacy of culturally specific religious primes are impacted not only by self-reported religiosity but also by religious socialization, identity, and ritual participation.

Perceptual cues associated with religion have been observed to affect decision-making across many behavioral domains [5–13] and their unconscious effects on ethical behavior have been studied extensively [1,14,15]. One particular behavioral domain studied in relation to religious cues is cheating. As an example, Aveyard [16] conducted two experiments investigating the effects of religious cues on ethical behavior in Middle Eastern participants. In the first experiment, participants unscrambled Arabic sentences with embedded religious or non-religious themes prior to taking a math test. The math test was incentivized, but unsupervised- so crucially, participants were able to dishonestly report their performance to earn more money. Aveyard [16] observed that participants' honesty was unaffected by their exposure to religious (or non-religious) anagrams. However, in a second experiment, listening to the Islamic call to prayer, *athan*, was observed to impact reporting in the unsupervised math test. Specifically, participants that listened to the Islamic call to prayer were more honest than the control participants who did not listen to the call to prayer. These results indicate that auditory religious primes can affect individual behavior. Further, Aveyard's findings suggest that individuals must have deep and natural associations with the sacred cue in order for it to change behavior.

Recently, however, religious priming has been the subject of a heated debate concerning the replicability of individual effects and the broader validity of this technique [1–3]. In a meta-analysis of 93 published and unpublished religious priming studies, religious priming was observed to have a consistent, small effect on the behavior of religious individuals [1]. These findings were contested as van Elk et al. [2] raised concerns that questionable research practices (QRPs) in social psychology confound meta-analysis results. They observed that religious priming effects are significant when Bayesian meta-analytic methods are used, but are non-significant when a Precision-Effect Testing–Precision-Effect Estimate with Standard Error (PET-PEESE) is used [2]. As these conflicting findings suggest, meta-analyses are not a panacea [17]; they are dependent upon the quality of the original data included and are subject to researchers' degrees of freedom involved with conducting analyses and interpreting the results [18]. PET-PEESE methodology, for instance, is sometimes unreliable when the true effect size is small and when the meta-analysis includes twenty or fewer studies, studies that have small sample sizes, or studies that exhibit a large degree of heterogeneity of effect sizes [17,19]. Taken together, while meta-analyses are useful and needed tools for research, they are often inconclusive and should be supplemented with replication efforts, including both direct and conceptual replications [2].

To determine the validity and generalizability of religious priming, improved experiment procedures and more diverse samples must be explored. Oftentimes, religious priming research has utilized dictator games to measure prosociality [6,7,20–24]. However, behavior in dictator games has been observed to lack global generalizability, as people from different cultures demonstrate unique norms for altruism when playing dictator games [25,26]. The method of religious priming can also influence results as recent work indicates that explicit religious primes (i.e., writing tasks) produce only small effects on prosociality, while implicit religious primes (i.e., anagrams) do not appear to influence responses to prosocial measures [22]. Furthermore, religious priming effects have primarily been studied using largely homogeneous, WEIRD (Western, Educated, Industrialized, Rich, and Democratic) samples, raising additional concerns about the generalizability of observed effects across other populations [1,27,28]. Indeed, a cross-cultural study of 15 small-scale societies found that religious priming inconsistently impacts people across cultural and religious contexts [29; see also 4,27]. Consequently, replication attempts should test novel, yet conceptually-relevant prosociality measures that seek to avoid the limitations of existing measures and should investigate if an observed effect replicates in the same sample (typically Western university students), but also across various cultural and religious settings, where an effect is claimed to have cross-cultural validity.

To that end, the current study attempts to replicate and extend the findings from Lang et al. [4], examining whether priming with instrumental religious music would decrease the rate of participants' dishonest behavior. Notably, this replication effort is distinct from what is sometimes referred to as a direct replication [30]. In this work, we adapt aspects of the methodology employed by Lang et al. [4] to investigate the conceptual consistency of their results. We designed this replication to provide results that would either increase or decrease the confidence in the interpretation of Lang et al. [4]. Therefore, this study utilizes the Open Science Framework (OSF) method of replication that, "reduces emphasis on operational characteristics of the study and increases emphasis on the interpretation of possible outcomes" [31]. (For a comparison of replication methodologies across disciplines, see Goméz and colleagues [30].) Below, we provide a succinct overview of Lang et al. [4]. Thereafter, we outline the unique aspects of this research and clarify the strategic motivations for modifying the procedure of Lang et al. [4]. Finally, in the Discussion, we detail how our adapted methodology may have limited our ability to replicate the findings observed in Lang et al [4].

Lang and colleagues [4]conducted a cross-cultural study on the effects of auditory cues on normative behavior using treatment with three musical stimuli (religious, secular, or white noise) prior to completing an arithmetic task designed to measure dishonest reporting for monetary reward (the Matrix Task, adapted from Mazar and colleagues [8]). Whereas they did not observe a main effect of musical treatment on dishonest reporting, the results revealed a Treatment*Religiosity interaction, such that participants self-reporting higher religiosity claimed less money upon hearing a religious musical track. Drawing from their results and previous research [16], Lang and colleagues [4] hypothesized that, "...instrumental music can serve as a reminder of normative behavior, but only for participants who previously formed an association between religion and specific music." Here, we re-examine this learned association hypothesis and test the impact of exposure to religious symbolism (music) on cheating behavior. As opposed to studying various forms of voluntary sharing and altruism that vary across cultures, we examine cheating as an anti-social behavior that is frequently and explicitly targeted by religious norms [32–34]. Critically, this study builds on previous efforts in four distinct ways.

First, to strengthen the link between musical stimuli and their effect on dishonest reporting, the musical tracks were played on-loop, throughout the entirety of the experimental task (rather than 2 minutes before the task as in Lang and colleagues [4]). Second, we use a

decision-making task that allows us to benchmark participants' self-reported earnings against factually correct answers over 200 total trials (the Dots Game, adapted from Gino and colleagues [35]). Compared to the previously used Matrix Task, the Dots Game provides a less biased measurement of cheating and allows the participant more opportunities to report dishonestly, strengthening the signal without increasing time requirements. In the Matrix Task employed by Lang et al. [4], the researchers were unable to determine participant accuracy in a specific trial. Rather, Lang et al. [4] measured an aggregate of self-reported 'correctly solved matrices' to make calculated assumptions about cheating behavior. In comparison, the Dots Game records accuracy in each trial and, therefore, enables the direct measurement of beneficial mistakes (cheating), which decreases the likelihood that genuine, erroneous participant judgments will impact our model. Third, we add a no-music condition to compare the impact of not listening to any sounds at all. Finally, we broaden the generalizability of this research by diversifying the sample, adding an East Asian (Japan) site to our sample in which the religious traditions represented are non-exclusive and focus more on practice (orthopraxy) than belief (orthodoxy) [36,37]. Given the population size of Japan (126.2 million: [38]) and the prevalence of orthopraxic religion in East Asia [39], adding a Japanese sample enables further examination of the generalizability of the hypothesized religious priming relationships. For a detailed summary of the differences between the design employed in this paper and the design used in Lang et al. [4], see S1 Table H in S1 File.

Data were collected from three culturally distinct samples that differ in intriguing ways in their general religiosity levels: the Czech Republic, Japan, and the USA. While the majority of people in Japan [40] and in the Czech Republic [41] indicate that religion is not important in their lives, the majority of people in the USA think religion is important in their lives [42]. Further, only a minority of people in the USA are not affiliated to any religion (16%), while the majority of people in the Czech Republic (76%) and in Japan (57%) report no affiliation [43,44]. Despite their high level of non-affiliation, however, over 40% of Japanese people indicated that they believe in god and a majority (62.7%) indicated they go to religious places at least once a year [45]. Japanese religious systems are also broadly non-exclusive and there is substantial overlap in the ritual and festival practices performed in both Shinto shrines and Buddhist temples [37,46–49]. Hence, differences between these sites should enable useful comparisons of the hypothesized effect, as they introduce variation in levels of ritual participation, religious belief, and the significance of religious affiliations. Critically, the inclusion of a Japanese sample enabled us to test for the effect of using musical performances drawn from a distinctive non-Western musical heritage, echoing Lang and colleagues' [4] choice to include Mauritian music.

Importantly, to date, there is only one published study that has investigated religious priming effects using a Japanese sample [50]. Miyatake and Higuchi [50] attempted a direct replication of Shariff and Norenzayan [6], utilizing identical methodology, and found that visual religious priming did not affect individuals' pro-sociality. As Miyatake and Higuchi [50] suggest, these results may have been due to their decision to use primes that relied on the Western, Christian references traditionally employed in religious priming techniques. Their suggestion that "if local religions and culture had been reflected in the religious primes, the results might have been different" [50] supported our decision to select culturally relevant auditory cues for each sample (see Materials).

To isolate the effects of religious music, we designed four conditions: religious, secular, white noise, and control (no music). Following Lang et al.'s [4] learned-association supposition and their results, we predicted there would be no main effect of condition on dishonest behavior, but we expected an interaction between religiosity and condition. Specifically, we expected that participants higher in self-reported religiosity would behave more honestly than their less-

religious peers but only when religious participants were listening to religious music. We also examined two supplementary hypotheses assuming: a) the moderating effects of affiliation to a religious organization that is congruent with the religious stimuli(religious affiliation) and b) the moderating effects of ritual participation frequency on the relationship between religious music and dishonest behavior.

The motivation for adding these supplementary hypotheses was to provide further nuance to previous findings by exploring the specific mechanisms that may facilitate the tentative effects of religious cues on normative behavior. The fact that Lang et al. [4] did not observe any main effect of condition on dishonest behavior and, consequently, that we do not expect such an effect in the present study supports the broader thesis that priming materials inherently connected to a specific cultural context will not affect people's behavior indiscriminately. That is, to the extent that priming effects rely on symbolic communication, they should be detected only in people who are able to access the symbol's conventional meaning and its connection to behavioral norms. Of course, some symbols include anthropomorphic characteristics that may exert additional effects on normative conduct. For instance, symbols with eyes may induce normative behavior because the presence of eyes, in general, indicates that one is being watched [13,51,52]. However, for most culturally specific cues(i.e., instrumental religious music), symbolic meaning and its association to moral norms must be learned and reinforced. In Lang et al. [4], the authors approximate individual reinforcement of religious music and its associated normative conduct by utilizing a concept of religiosity that subsumes dimensions such as religious belief, practice, experience, and commitment [53]. While this broad measure of religiosity is a useful and easy to use approximation, it does not afford for the precise estimation of an individual's commitment to and understanding of culturally specific religious symbols.

Indeed, being religious or spiritual does not guarantee that one will understand the meaning of a symbol and its connection to the normative structure of a religious system. In multi-religious contexts (USA and Japan in our sample) or in contexts where people declare to be religious/spiritual but unaffiliated (Japan and the Czech Republic in our sample), people may believe in various supernatural agents and take part in many religious activities that are not directly connected to the religious system from which we sampled our priming material. In other words, participants who do not self-affiliate with religious organizations that practice the specific tradition of our religious stimuli may not have learned the conventional meaning and associations of the specific cue used in the present study (despite self-reporting high religiosity). Compared to religiosity, affiliation may serve as a more fine-grained predictor of the learned association between a religious cue and normative behaviors. To examine this idea in greater detail, we measured affiliation and tested its interaction with our treatment, anticipating that religious affiliation would enhance the effects of religious music on ethical behavior.

Furthermore, in many Western religious systems (i.e., Protestantism), religiosity is often associated with personal belief [54–56]. However, in other more orthopraxic oriented religious systems, the dominant dimension of religiosity may be ritual behavior. Together, collapsing orthopraxic and orthodoxic perspectives under the measure of religiosity obfuscates clear distinctions between the effects of practice and belief on behavior [54,57]. Critically, Lang et al. [4] suggested that it is through communal rituals, rather than belief itself, that the conventional association between a sacred symbol and norms are established and perpetuated. Relatedly, previous findings have observed that increased frequency of ritual behavior facilitates favorable treatment of other co-religionists in real-life and in laboratory contexts, and across various economic games [4,12,58–60]. Music is central to many religious rituals [61,62], serving several functions like coordination and synchronization [10,63,64], but also creating lasting associations between ritual context and normative conduct. Importantly, Lang and colleagues [4]

observed a relationship between frequency of ritual participation and the effectiveness of an auditory religious prime on moral behavior. Thus, to provide a more nuanced investigation of this issue, we also included a measure of ritual participation frequency and interacted this measure with our manipulation, expecting that frequent ritual participation would strengthen the effects of religious music on honest behavior.

## Materials and methods

### Participants

Data were collected at universities across three sites: the USA, the Czech Republic, and Japan. A total of 460 (228 females) adults were randomly assigned to one of four conditions: religious, secular, white noise, or control (no sound). Due to self-reported suspicion on the goals of the experiment and previous experience with studies using the Dots Game, a total of 52 participants were excluded from the analysis. In support of this decision, supplementary analyses of the incorrectly claimed higher-paying sides in the Dots game showed that the excluded participants had 5.5 higher odds of incorrectly claiming all money from the higher-paying side (100% claimed) compared to participants in the included sample (see S1 Table G in S1 File for the analysis of the full sample). Five additional participants had missing data on crucial variables such as sex and age. Data included in the analyses therefore comprise: 123 American participants ($M_{age}$ = 25.5, SD = 9.8), 128Czech participants ($M_{age}$ = 24.4, SD = 3.4), and 157 Japanese participants ($M_{age}$ = 19.8, SD = 0.9). Within the four experimental conditions, there were:100 participants in the control condition, 103 participants in the white-noise condition, 103 participants in the secular condition, and 102 participants in the religious condition. Based on the increase of 0.023 in $R^2$ for the interaction between condition and religiosity found in Lang et al. [4], this sample size should give us 0.85 power to detect the same interaction at alpha = 0.05 (calculated in G*Power).

Experiments were conducted in laboratory settings containing tables, chairs, and computers with headphones. Participants were seated in cubicles such that only the content of their own screen was visible. Experimental materials, informed consent forms, and scripts were translated into local languages (Czech and Japanese) from English. The institutional review boards of Duke University, Hokkaido University, and Masaryk University approved this research. All participants provided an informed consent before taking part in the experiment.

### Materials

The Dots Game is designed to measure participants' willingness to cheat for real monetary rewards (adapted from Gino et al. [35]). The Dots Game is a digital task consisting of 200 trials in which dots quickly appear and disappear on a vertically bisected screen. In order to ensure comprehension, all participants completed 10 practice trials before starting the 200 trials that earned compensation. After each trial, participants were asked to indicate the side of the screen (left or right) that had contained more dots. There was no time limit for participants to make their decisions.

The total number of dots presented in each trial summed to 22 and a minimum of eight dots was randomly presented on the left or right side in every trial. Of the 200 payment trials, 120 displayed more dots on the left side while only 80 trials displayed more dots on the higher paying, right side. During the task, earnings accumulated and were displayed at the top of each participant's screen as selections were made. Although participants were instructed to be as accurate as possible, accuracy does not affect payment in the Dots Game.

It was explained to participants that detecting more dots on the left side of the screen is easier and that the payment for right-side selections would therefore be worth more than left-side

selections. Specifically, indicating that the left side of the screen contained more dots always earned the participant $0.005 USD while indicating that the right side of the screen contained more dots always earned the participant $0.05 USD. To measure cheating, we do not look to the total number of higher paying selections made [35]; rather, we limit response bias by observing the percentage of inaccurately claimed higher-paying sides [65]. In Japan and the Czech Republic, compensation was paid in local currency and approximately equaled USD amounts ($0.05 and $0.005). Therefore, a maximal cheater could earn $10 and a completely honest and accurate player would earn $4.60. Musical tracks (religious, secular, white noise) played on-loop were added to the Dots Game for the purposes of this experiment.

Musical tracks were pre-tested using participant pools specific to each site (online with Lancers in Japan and Amazon's Mechanical Turk in the USA, and with a student population in the Czech Republic). At each site, we compared eight 2-minute musical tracks across their musical characteristics, including tempo, affect, and impact. See S1 Table I in S1 File for the musical tracks tested in each population. The songs selected for the USA and CzR samples were identical to the ones used in Lang and colleagues [4]. For the USA sample, Johan Sebastian Bach's *Jesu joy of man's desiring* was chosen as the religious track, while Bach's *Sleepers awake* was chosen as the secular track. In the Czech Republic, Bach's *Ave Maria* (Gounod's interpretation) was used as the religious music, while Tchaikovsky's *Romance for piano in F Minor, Op. 5* was selected for the secular music. A *Gagaku*(雅楽) musical track was selected as the religious stimuli for the Japan sample. *Gagaku* music is a form of traditional classical music performed by an orchestra and usually features the distinctive sound of a traditional mouth organ, referred to as *Shō*(笙). *Gagaku* has been described as the world's oldest orchestral music and is associated with Shinto ritual performances and ceremonies conducted at the imperial court [66,67]. For the Japanese secular music condition, music performed on a *koto* (箏), a traditional Japanese stringed instrument, was selected. Furthermore, in order to avoid any associations with traditional ritual or religious settings, a *koto* performance of a more contemporary arrangement was selected [68]. Across all sites, the selected religious and secular musical tracks were instrumental only and included no vocal elements. They differed in perceived sacredness but were similar in tempo and affect (See 'Manipulation Check'). Finally, the white noise comprised a loop of white noise played through headphones at all sites, while the control condition comprised just silence.

Surveys were administered after completion of the Dots Game to assess religiosity (0 –Not religious at all, 4 –Very religious/spiritual person), ritual attendance frequency (0 –Never, 6 – More than once per week), religious organization affiliation (i.e., church), and religious tradition participants were affiliated with. Participants in the music conditions (religious, secular, and white noise) rated how secular/religious and profane/sacred the sound was on a 7-point Likert scale (1 = Secular, 7 = Religious; 1 = Profane, 7 = Sacred). Additionally, music condition participants used a 5-point Likert scale (1 = Not at all, 5 = Extremely) to rate the extent to which the song they heard was: sad, fast, boring, pleasant, happy, irritating, slow, exciting, deep, interesting, distressing, powerful, relaxing, and distracting. Recognition of the musical track (Yes, No) was also assessed for music conditions' participants. All participants were asked about the perceived difficulty of the task (1 –Very easy, 5 –Very difficult), as well as their age and gender. Given the Dots Game was developed and studied locally, US participants also indicated their involvement in previous research using the Dots Game (Yes or No).

## Procedure

Participants were first randomly assigned to one of four conditions: religious music, secular music, white noise, or control (no music). Participants were informed that the research was studying decision-making and at each site, local research assistants facilitated the experiment.

Upon arrival to the lab, participants were seated in front of a tablet or a computer and could see their screen only. First, participants read instructions for the Dots Game where it was explained that the game consisted of 200 trials in which dots would temporarily flash onto computer screens. For the duration of the game, the computer screens were divided into two vertical halves (left and right) and, after each trial, participants were asked to accurately determine which side of their screen contained the majority of the dots that had appeared by pressing either the 'M' key (to indicate right) or the 'Z' key (to indicate left) on a computer keyboard. Dots remained visible on the computer screen for only one second before participants were prompted to make their selection. As participants made their selections, earnings accumulated and were displayed at the top of each participant's screen. On average, it took participants six minutes and four seconds (SD = 1.15 minutes) to finish the Dots Game.

All participants were instructed to wear the headphones at their computer for the entire duration of the game. Control participants played the Dots Game without music; all other participants played the Dots Game while listening to their site-specific and randomly assigned musical track on-loop. The research assistant was available in an adjacent room to provide any necessary assistance. After the game was over, participants completed a post-study questionnaire (see Materials for overview, Supporting Information for detailed review) and received the reward amount they had earned in the Dots Game. Completing the Dots Game and survey took participants no more than 30 minutes.

## Results

All data were analyzed in R (version 3.4.3, R Core Team 2017). We first constructed an Ordinary Least-Square Regression (OLS) model with treatment as a factor variable, investigating the main effects of musical condition on the percentage of dishonestly claimed earnings. We set the religious condition as a reference category to compare its effects with various controls (secular music, white noise, control), while holding the effects of age, gender, and site constant as simple fixed effects. Note that the USA was set as a reference category for the site factor variable; however, this selection was arbitrary and did not affect any of the main estimates of interest. Next, three interaction OLS models were constructed, looking at the moderating effects of religiosity, ritual participation, and religious affiliation on the relationship between the treatment and dishonest behavior. Religious affiliation was determined using religious organization affiliation and religious identity responses. Specifically, a participant was considered affiliated if they belonged to a religious organization and self-identified with the religion associated with the religious stimulus at each site (i.e., Christianity in the Czech Republic and US, Shinto in Japan). Note that while Lang and colleagues [4] used beta regression to model the percentage data in their previous analyses [69], the current results from OLS regressions are qualitatively similar to the results of beta regression; hence we opted for simpler models. The results from beta regression models are reported in the, S1 Table D in S1 File. Likewise, since our participants were nested within sites, it would be more appropriate to use linear mixed models to investigate our main hypotheses. However, given that there are only three categories in our nesting variable, estimating individual site intercepts from the partially pooled data did not yield qualitatively different results compared to using sites as simple fixed effects (see S1 Table E in S1 File). Additionally, we adjusted our models to account for: the perceived difficulty of the dots task, the difference between the average trial completion time, and the completion times for trials where participants cheated. In the final robustness check, we hold constant the ratings of musical stimuli to ensure that the observed effects were not caused by differences between the stimuli's perceived affect, tempo, or impact (see S1 Table F in S1 File).

## Manipulation check

Perceived sacredness was observed to be significantly different between musical conditions [F(2,299) = 53.8]. Across all sites, the religious track received significantly higher ratings of sacredness than did the secular($\beta$ = -0.85; 95% CI = [-1.21, -0.49]) or white-noise tracks ($\beta$ = -1.91; 95% CI = [-2.27, -1.54] see Table 1 for descriptive statistics). Similar results were obtained with the secular/religious measure [F(2,300) = 40.34], showing higher religiosity ratings of the religious song compared to the secular ($\beta$ = -1.32; 95% CI = [-1.69, -0.95]) or white-noise tracks ($\beta$ = -1.60; 95% CI = [-1.97, -1.23]).

## Dishonest behavior

Dishonesty in the Dots Game was observed by measuring the proportion of inaccurate higher paying (right side) selections made. This dishonesty metric was calculated by dividing the number of times a participant inaccurately indicated that the higher paying side contained more dots by the number of trials (120) in which the lower paying (left side) truly contained more dots. Participants across all sites earned an average of $5.86 (SD = $1.52), indicating right side incorrectly on average in 27.39% of trials (SD = 29.21%). Interestingly, the average rates of dishonest reporting differed between our sites: while in the Czech Republic, participants claimed on average 11.69% (SD = 13.72%) incorrectly, in Japan and the USA the rates were as high as 29.88% (SD = 28.43%) and 40.56% (SD = 34.28%), respectively (see Fig 1B).

Looking at the distribution of dishonest reporting across our musical treatment, we observed that the control condition had the lowest amount of cheating, followed by the religious, white-noise, and secular conditions (see Table 1 and Fig 1A). However, these raw results ignore the hierarchical structure of our data where participants are nested within sites. Hence, to examine the effects of our treatment on dishonest reporting more rigorously, we regressed the incorrectly claimed right sides on our musical treatment, holding the site-specific mean levels of dishonest reporting constant. This regression model revealed that there were no substantial differences between the religious and the secular ($\beta$ = 3.45; 95% CI = [-4.02, 10.91]), white-noise ($\beta$ = 3.21; 95% CI = [-4.21, 10.62]), or control conditions ($\beta$ = -4.91; 95% CI = [-12.37, 2.55]; see Table 2). The inability to find differences between conditions replicates Lang and colleagues' [4] previous finding.

Following the absence of treatment main-effect, we tested three moderator models, investigating the role of self-declared religiosity, ritual frequency, and religious affiliation. First, we did not observe an interaction between condition and self-reported religiosity. While an increase in self-declared religiosity predicted decrease in the proportion of incorrectly reported

**Table 1. Descriptive statistics of aggregate unethical behavior and post-experiment ratings of musical stimuli.**

|  | Religious | | | | Secular | | | | White Noise | | | | Control | | | |
|---|---|---|---|---|---|---|---|---|---|---|---|---|---|---|---|---|
|  | (*n* = 102) | | | | (*n* = 103) | | | | (*n* = 103) | | | | (*n* = 100) | | | |
|  | **M** | **SD** | **CI** | *d* | **M** | **SD** | **CI** | *d* | **M** | **SD** | **CI** | *d* | **M** | **S** | **CI** | *d* |
| % Claimed | 27.33 | 29.92 | [21.52,33.14] | - | 30.32 | 30.98 | [24.34,36.30] | 0.10 | 29.40 | 31.09 | [23.40,35.40] | 0.07 | 22.38 | 23.89 | [17.69,27.06] | 0.18 |
| Sacredness | 5.17 | 1.39 | [4.90, 5.44] | - | 4.32 | 1.35 | [4.06, 4.58] | 0.62 | 3.26 | 1.19 | [3.04, 3.49] | 1.48 | - | - | - | - |
| Negativity | 1.82 | 0.65 | [1.70, 1.95] | - | 1.64 | 0.51 | [1.55, 1.74] | 0.31 | 2.58 | 0.96 | [2.39, 2.76] | 0.93 | - | - | - | - |
| Positivity | 2.54 | 0.85 | [2.37, 2.70] | - | 2.72 | 0.78 | [2.56, 2.87] | 0.22 | 1.38 | 0.62 | [1.26, 1.50] | 1.56 | - | - | - | - |
| Tempo | 2.60 | 0.81 | [2.45, 2.76] | - | 2.78 | 0.82 | [2.62, 2.94] | 0.22 | 3.99 | 0.72 | [3.85, 4.13] | 1.80 | - | - | - | - |
| Impact | 2.90 | 1.10 | [2.70, 3.10] | - | 2.73 | 1.13 | [2.51, 2.95] | 0.16 | 1.76 | 0.93 | [1.58, 1.94] | 1.13 | - | - | - | - |

M = Mean; SD = Standard Deviation; CI = 95% Confidence intervals. Cohen's d represents the effect size of comparisons between musical conditions.

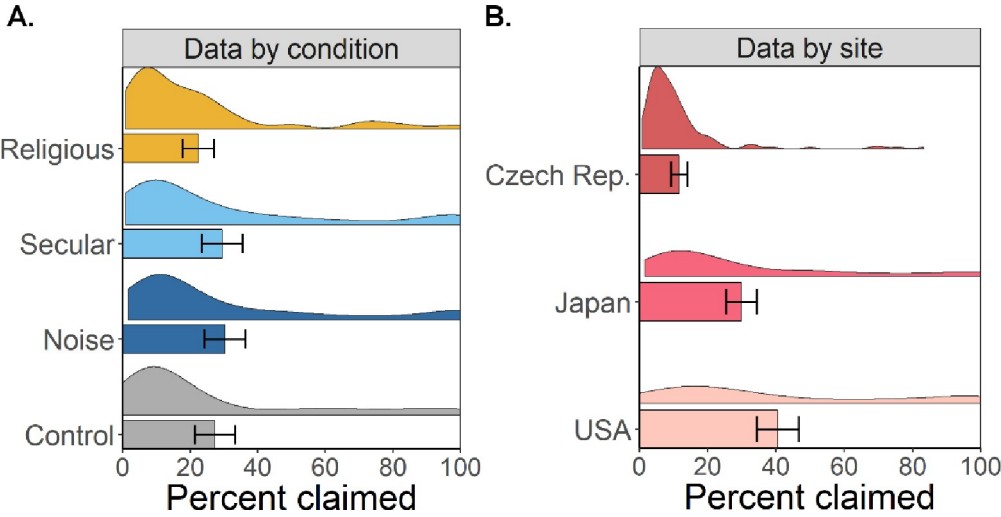

**Fig 1.** Mean values for dishonestly claimed earnings with 95% CIs divided by condition (A) and site (B). Each level displays a bar with 95% CIs and a density plot.

right sides in the religious music condition ($\beta$ = -2.12), this decrease was imprecisely estimated and the 95% CI crossed zero (-6.77, 2.53). Furthermore, this religiosity coefficient was not substantially different from coefficients in the secular ($\beta_{difference}$ = 2.97; 95% CI = [-3.40, 9.35]), white-noise ($\beta_{difference}$ = -1.40; 95% CI = [-7.94, 5.14]), or control conditions ($\beta_{difference}$ = 0.82; 95% CI = [-5.49, 7.13]; see Fig 2A). For site-specific results, see S1 Table A in S1 File.

Following the effects of ritual participation found in Lang et al.[4], we built a second moderator model measuring the effect of the ritual attendance frequency and its interaction with treatment. An increase of one on the ritual frequency scale (0 –Never, 6 –More than once per week) predicted a decrease of roughly 4 percentage points in dishonestly claimed compensation in the religious condition ($\beta$ = -4.19; 95% CI = [-7.34, -1.03]). Importantly, the slope of the ritual attendance differed between conditions, showing that ritual attendance decreased the ratio of incorrectly claimed right sides only to a small extent in the secular condition ($\beta_{difference}$ = 3.46, 95% CI = [-1.01,7.94]), and had no effect in the noise ($\beta_{difference}$ = 4.49, 95% CI = [-0.10, 9.08]) and control conditions ($\beta_{difference}$ = 3.89; 95% CI = [-0.48,8.26]). See also Table 2 and Fig 2B and S1 Table B in S1 File for site-specific results.

Finally, in the third moderator model, we analyzed the effects of self-reported affiliation matching the specific religious stimulus at each site(binary yes/no variable) and its interactions with the musical treatment. Religious affiliation had a strong negative relationship with dishonest reporting in the religious condition, predicting roughly 26 percentage points lower number of unfairly claimed compensations ($\beta$ = -26.09; 95% CI = [-40.12, -12.06]). Importantly, we observed a significant Condition*Affiliation interaction: the effect of religious affiliation was much weaker in all three remaining conditions (Secular: $\beta_{difference}$ = 23.78, 95% CI = [5.64, 42.90]; Noise: $\beta_{difference}$ = 21.84, 95% CI = [2.76, 40.92]; Control: $\beta_{difference}$ = 24.94, 95% CI = [5.97, 43.91]; see Table 2 and Fig 2C). These findings indicate that participants who affiliated with a religious tradition matching our stimuli were less dishonest when listening to the religious song, but affiliation had no effect in the remaining conditions (see S1 Table C in S1 File for site-specific results). See also, S1 Table D in S1 File for a robustness check of these results using the Beta regression and S1 Table E in S1 File for using linear mixed models (these robustness checks support our findings obtained with simpler models reported here).

**Table 2. Estimates with 95% CIs from Ordinal Least Squares regressions for the percentage of higher-paying side (right) claimed as having more dots.**

|  | M1: Baseline | M2: Religiosity | M3: Ritual frq. | M4: Affiliation |
|---|---|---|---|---|
| Intercept | 38.06*** | 42.08*** | 46.61*** | 44.61*** |
|  | (31.09, 45.02) | (31.37, 52.80) | (36.86, 56.35) | (36.75, 52.47) |
| Secular | 3.45 | -1.34 | -2.70 | -0.78 |
|  | (-4.02, 10.91) | (-14.26, 11.57) | (-13.84, 8.44) | (-9.31, 7.76) |
| Noise | 3.21 | 5.81 | -5.82 | -0.76 |
|  | (-4.21, 10.62) | (-7.31, 18.93) | (-16.85, 5.22) | (-9.04, 7.52) |
| Control | -4.91 | -6.12 | -11.41* | -9.52* |
|  | (-12.37, 2.55) | (-18.88, 6.63) | (-22.13, -0.68) | (-17.88, -1.17) |
| Sex | 2.18 | 1.93 | 2.51 | 2.65 |
|  | (-3.24, 7.60) | (-3.62, 7.48) | (-3.04, 8.07) | (-2.82, 8.13) |
| Age | 0.49* | 0.47† | 0.52* | 0.48* |
|  | (0.03, 0.96) | (-0.004, 0.94) | (0.06, 0.99) | (0.02, 0.94) |
| Site: Czech Rep. | -28.56*** | -28.90*** | -29.74*** | -30.86*** |
|  | (-35.27, -21.85) | (-35.84, -21.97) | (-36.74, -22.73) | (-37.84, -23.88) |
| Site: Japan | -8.14* | -9.23* | -10.26* | -11.32** |
|  | (-15.23, -1.06) | (-16.67, -1.79) | (-18.03, -2.49) | (-18.72, -3.92) |
| Moderator | - | -2.12 | -4.19** | -26.09*** |
|  | - | (-6.77, 2.53) | (-7.34, -1.03) | (-40.12, -12.06) |
| Secular*Moderator | - | 2.97 | 3.46 | 23.83* |
|  | - | (-3.40, 9.35) | (-1.01, 7.94) | (5.64, 42.01) |
| Noise*Moderator | - | -1.40 | 4.49† | 21.84* |
|  | - | (-7.94, 5.14) | (-0.10, 9.08) | (2.76, 40.92) |
| Control*Moderator | - | 0.82 | 3.89† | 24.94* |
|  | - | (-5.49, 7.13) | (-0.48, 8.26) | (5.97, 43.91) |
| Observations | 403 | 395 | 384 | 397 |

Moderator is either religiosity, ritual frequency, or religious affiliation, see model names. The condition*moderator interactions represent the estimated differences between the slope of the moderator in the religious condition and moderator slopes in the other conditions.

†$p < 0.1$

*$p < .05$

**$p < .01$

***$p < .001$.

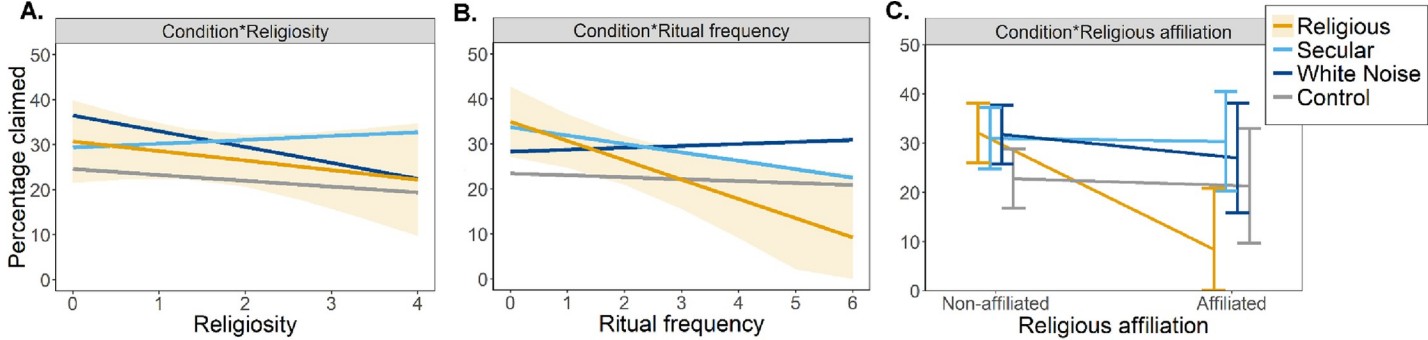

**Fig 2. Interaction plots with predicted values for dishonestly claimed earnings and 95% CIs. A.** The regression slope of religiosity in the religious condition did not differ from the other conditions. **B.** Ritual frequency predicted decreased cheating, and the regression slope differed from other condition (albeit the 95% CI crossed zero for the difference between the religious and secular conditions). **C.** Self-declared affiliation to religious organization congruent with our stimuli predicted decreased cheating in the religious condition but not in the other conditions. We display 95% CIs only for the religious condition for easier reading. All CIs are displayed in Table 2.

As a final robustness check, we also adjusted our models for the mean completion time of the dots task, perceived difficulty of the task, and musical characteristics of our stimuli (see S1 Table F in S1 File). Across all models, we observed that the rates of dishonest behavior were predicted by faster completion times on the dishonestly reported trials. That is, the less time participants dedicated to decision making, the more likely they were to report dishonestly because correct answers required deliberately counting the dots and making sure one selected the correct answer. This finding is congruent with perceived difficulty of the task, which negatively predicts the number of incorrectly reported dots. Notably, these findings can be explained by dishonest participants' willingness to pre-determine the higher compensation choice (hold down the 'M' key), which would naturally decrease participation time and task difficulty. After adjusting our models for these variables, the moderating effects of ritual participation frequency and religious affiliation remained stable. Furthermore, we assessed whether the reported results hold even after the models accounted for the musical characteristics of our stimuli: tempo, influence, positivity, and negativity (see S1 Table F in S1 File). While perceived negativity of the played track predicted dishonesty, we still observed the effect of ritual and religious affiliation as well as the interactions of Condition*Affiliation and Condition*Ritual, albeit we could not assess the effects of musical characteristics in the control condition due to the fact this condition had no musical stimulus. These findings indicate that the religious condition tracks did not affect cheating in the Dots Game due to their musical characteristic ratings; rather, participants who affiliated with a religious organization and participated in rituals were uniquely affected by the sacred music primes and, in turn, played less dishonestly than unaffiliated participants.

## Discussion

In the present study, we conducted a cross-cultural replication and extension of religious priming research by Lang et al. [4]. Specifically, we tested the hypothesis that auditory religious cues decrease unethical behavior in cheating games compared to other auditory cues (secular, white noise, and control). We collected data on university populations from three countries with distinct religious and cultural norms: the Czech Republic, Japan, and the United States of America. All participants played a cheating game, the Dots Game, during which they listened to a musical track and had an opportunity to maximize their earnings by playing dishonestly. There was no main effect of musical treatment on cheating behavior. Further, we were unable to replicate the interaction effect of Condition*Religiosity observed by Lang et al. [4]. However, we did observe interactions between condition and religious affiliation and condition and ritual participation. Auditory religious cues were found to decrease dishonest reporting for participants frequently attending religious services and for those affiliated with a religious organization matching our religious stimuli.

Taken together, our results generally provide support for Lang and colleagues' [4] observation that exposure to religious music is not enough to prime honest behavior in all contexts. Notably, our findings provide additional evidence for a mechanism they proposed; specifically, an entrenched association between music and the religious values it reinforces is required for the activation of normative behavior. In their paper, Lang & colleagues [4] observed that situational religious factors, such as reported religiosity and ritual participation, played a role in the activation of religious cues and facilitation of their effects on ethical behavior. Although we did not observe the Condition*Religiosity interaction in the present research, we did observe similar interactions indicating that individual religious characteristics play a role in how people react to religious auditory cues.

In order to understand why we were unable to replicate the Religiosity*Treatment interaction, it is important to consider two points. First, compared to the Mauritian sample in Lang et al. [4], our Japanese sample included greater religious diversity. All Mauritian participants were Hindu in Lang et al. [4], hence the selected religious music was generally familiar and associated with their affiliated tradition. In our Japanese sample, however, participants from a variety of religions were represented (Christianity, Buddhism, Shinto, Judaism, atheism, unknown, and other). While *Gagaku* music is sometimes performed as part of Buddhist religious ceremonies, it is most strongly associated with Shintoism. Therefore, it is possible that for some of our participants this music did not cue normative behavior typical for the targeted religious affiliation. Second, religion in Japan, as elsewhere in Asia, is typically regarded as non-exclusive and has been described as having a 'practical' orientation wherein adherence to religious beliefs is seen as of secondary importance to ritual practices [37,39,70], potentially confounding the measure of religiosity. For instance, Kavanagh and Jong [39], recently demonstrated that while only 10% of 1,000 Japanese respondents self-identified as religious, 34% in the same sample identified as Buddhist, 5% as Shinto, and 33% endorsed the existence of God.

The increased religious diversity of our sample implies that religiosity in the current paper has a distinct meaning as it is understood within the context of an individual's religious background. Indeed, the site-specific analyses of the Condition*Religiosity interaction (see S1 Table A in S1 File) revealed that in Japan, religiosity had the smallest effect on dishonest behavior in the religious condition (albeit these coefficients were imprecisely estimated). This finding is in line with similar research showing that in multi-religious societies, cross-religious symbols may reduce trust and cooperation [71, but see also 72].

Secondly, the relationship between affiliation, ritual participation, and religiosity is complex. It is entirely possible for someone to be religious/spiritual and, at the same time, be unaffiliated with a specific religion or religious organization. In the USA, for instance, where religiosity is typically understood as affiliation to a specific church [73], more than a quarter (27%) of adult respondents indicated they were spiritual, but not religious [42]. However, if a person is spiritual without being affiliated to a specific religious institution, they may lack the necessary associations with the sacred auditory cues we selected and therefore could be less, or entirely, unaffected by our stimulus. To make the relationship clearer, we coded affiliation to specific religious organizations and demonstrated that membership in the matching tradition had an effect. We conjecture that this is due to people affiliated with a religious organization having more exposure to sacred music in meaningful contexts than unaffiliated peers and consequently, affiliated persons are likely to have stronger associations to the moral implications of religious auditory cues than the unaffiliated. In support of this, research on exposure and learning has demonstrated that a person's affective response to a stimulus can change over time [74]. In addition to repeated exposure, the meaningfulness of a stimulus can affect a person's response to it [75]. With repeated exposure to meaningful stimuli, people are able to create associations and have the ability to strengthen their judgments about the stimuli and its propositions [76].

It is also important to consider the fact that unaffiliated religious people often have differing beliefs about the connection between specific religions, belief in God(s), and morality [77]. The majority of religiously affiliated people in the USA (55%) agree that the belief in God is necessary to be moral; conversely, however, of all adults—affiliated and unaffiliated—the majority in the Czech Republic (78%), Japan (55%), and in the USA (56%), do not think it is necessary to believe in god to be moral [44,78]. Based on previous literature and on the findings of the present research, we suggest that religious auditory stimuli cue ethical behavior only for participants who believe that their affiliated religion is inherently linked to morality.

This interpretation is congruent with the results from the regression model of the interaction between condition and the frequency of ritual attendance, where ritual attendance has the

largest negative effect on dishonest reporting in the religious condition. Through repetitive reminders of sacred cues during religious ceremonies, the association between a specific sacred cue and the moral doctrine of a specific religion is strengthened and the cue is emotionally charged with special significance [61]. This in turn may result in a larger influence over participants' behavior being exerted upon perception of the cue [79,80]. Importantly, collective rituals are also a public venue for communicating commitment to supernatural agents and the norms they impose on believers [80–82]. However, the commitment is not signaled only to other believers but also to oneself as a form of auto-signaling reassuring participants about their beliefs in supernatural agents [58,83,84]. Hearing religious music may be a subconscious reminder of participation in rituals, strengthening the auto-communicated commitment signals. While the effect of ritual participation was weaker compared to religious affiliation (similar to the findings of Lang et al. [4]), we still detected a signal supporting this interpretation.

It is important to note that our analyses do not take into account what type of rituals our participants attend as well as the frequency with which our religious stimuli are played during specific religious ceremonies; hence, the signal is noisy. A larger, high-powered sample allowing for testing a three-way interaction between treatment, religiosity, and affiliation to specific religious organization may solve this issue in future studies. Furthermore, our sample consisted predominantly of university students. While the sample was culturally and religiously diverse, it generally lacked diversity in age or employment status. Moreover, due to specific lab guidelines, participants in the Czech sample received an additional reward of points that were redeemable for a course credit, which may have partially decreased their motivation to report dishonestly for monetary compensation. Indeed, this motivation may have acted as a boundary condition for religious priming across samples. As previously noted by other researchers using priming techniques [23,29], there may have been little room to observe religious priming effects if the motivation to be dishonest was relatively low in the Czech Republic. To avoid these problems, cross-cultural researchers should be careful to align laboratory incentive structures across samples. Moreover, future studies on priming with religious music should consider sampling from more populations with cultural and demographic diversity that would allow for the assessment of between-site differences in the hypothesized effects (see S1 Tables A-C in S1 File). Likewise, since stimulus selection can influence Type I error rates if stimuli are not representative of their theoretical construct and are treated as fixed-factors [85], the selection of locally salient religious stimuli should be made more robust by including at least three different stimuli at each site. Future researchers should also explicitly control for stimulus variation by employing mixed models that treat both participants and stimuli as random factors [86].

Additionally, it is useful to consider the limitations of single-item scales when interpreting our results [87]. The reliability of the religiosity measure we employed was not clearly established in this research. This measure contained only a single-item and was not pre-tested between sites prior to experimentation [87]. Therefore, it may have been the case that religiosity was understood and expressed idiosyncratically within each of our cross-cultural samples. Furthermore, the religiosity scale was asymmetrical as the word "spiritual" was not included at both ends of the scale (see Questionnaire materials in S2 File). This scale asymmetry was consistent across all sites and was due to an error in implementing the materials of Lang et al [4]. Together, the lack of pre-tested validity of our single-item measure as well as the scale's asymmetry, may have limited our ability to replicate the primary Condition*Religiosity interaction observed in Lang et al [4]. To explicitly address such concerns, future researchers should investigate the reliability of different religiosity scales across cultures, peoples, and religions.

Finally, it is worthwhile to discuss a limitation of the Dots Game procedure. The Dots Game has been utilized to measure ethical decision-making preferences for nearly a decade

[35] (See Materials for a review of the procedure). In this research, we use the Dots Game to observe cheating behavior as it provides a less biased- and more granular- measurement than the Matrix task employed by Lang and colleagues [4]). Furthermore, we investigated religious priming effects on honesty because normative regulations of altruism and sharing may vary across cultures substantially more compared to norms regulating cheating [27,29]. Despite the benefits of using the Dots Game (see in-depth discussion in Introduction), our results may be influenced by signal detection biases (for a review of signal detection research, see [88–90]). Due to the compensation scheme, participants may have preferred the higher-paying (right) side and therefore, may have had an unconscious bias to detect more dots on the higher-paying side.

Indeed, research has shown that Dots Game participants attend more to the higher-paying side in incentivized Dots Games [65]. Unconscious attentional bias may increase ambiguity, making it less clear to participants when their choices are inaccurate. Thus, cheating behavior may have been facilitated by an interaction of signal detection bias with various factors(e.g., noise, perceived difficulty or distraction). Greater familiarity with a musical track, for example, could have increased participants' bias for detecting higher-paying side dots, which in turn may have increased participants' likelihood to report inaccurate higher-paying selections. However, given the potential for third variable problems, we adjusted our models for perceived difficulty and musical ratings (See Supporting Information for an in-depth review). Furthermore, it is unlikely that unconscious signal detection biases can fully explain the cheating behavior we observed in this paper, as other findings indicate that Dots Game participants are at least partially conscious of their unethical behavior [65].

We recommend that future research extends the religious priming literature by exploring methods that will provide less biased cheating data. Specifically, future studies should test the religious priming effects on a broader spectrum of samples, including horticulturalists and pastoralists from small-scale societies [29]. Moreover, we encourage future researchers to independently investigate if the differences observed between our samples replicate. For instance, cross-cultural researchers could attempt to replicate and extend the understanding of the uniquely low cheating rates observed in our Czech sample, as well as the depressed Condition*Religiosity interaction coefficient observed in the Japan model. Additional, experiments could test religious priming effects on cheating behavior across a variety of cheating tasks. Finally, we encourage healthy science practices and invite researchers to improve the generalizability of our findings by iterating on our replication with pre-registered designs. In support of such future endeavors, we have made our study materials and data available for access on the Open Science Framework (https://osf.io/k4dt8).

## Conclusion

In summary, we conceptually replicated findings in the religious priming literature that indicate sacred cues affect individual ethical behavior and support a learned association hypothesis [4]. Although we did not find evidence for the expected interaction effect between religiosity and musical primes, we did observe an interaction effect between ritual participation and musical prime and between religious affiliation and musical prime. More specifically, sacred auditory cues were found to affect ethical behavior for individuals who attend religious rituals or were affiliated with a religious organization that practices the tradition associated with the relevant musical cues. These indirect priming effects were congruent with Lang and colleagues [4], even though the current paper extended the research design by utilizing a decision-making task better able to detect cheating (the Dots Game) and by including a non-Western site with greater religious diversity and an orthopraxic religious orientation (Japan).

It is our hope that the current research will inspire others to conduct replications and further examinations of religious priming effects, especially with understudied populations. Indeed, replications have been identified as a solution to the reproducibility crisis in the social sciences and may one day end debates within religious priming research [13,15,16].

## Supporting information

**S1 File.**
(DOCX)

**S2 File. Post-study questionnaire materials, translated from English into Czech and Japanese for the Czech Republic and Japan sites, respectively.**
(DOCX)

**S1 Data.**
(ZIP)

**S2 Data.**
(ZIP)

## Acknowledgments

We would like to thank Dhrumil Patel and Merve Akbas for advising on the Dots Game software. Jan Horský, Monika Bystroňová and František Brázda for being invaluable research assistants, and Eva Kundtová Klocová and HUME Lab Experimental Humanities Laboratory, Faculty of Arts, Masaryk University, for exceptional research support.

## Author Contributions

**Conceptualization:** Aaron D. Nichols, Martin Lang, Panagiotis Mitkidis.

**Data curation:** Aaron D. Nichols, Martin Lang, Christopher Kavanagh, Radek Kundt, Junko Yamada.

**Formal analysis:** Aaron D. Nichols, Martin Lang.

**Funding acquisition:** Martin Lang, Radek Kundt.

**Investigation:** Aaron D. Nichols, Junko Yamada.

**Methodology:** Aaron D. Nichols, Martin Lang.

**Project administration:** Aaron D. Nichols.

**Resources:** Dan Ariely.

**Supervision:** Dan Ariely, Panagiotis Mitkidis.

**Visualization:** Martin Lang.

**Writing – original draft:** Aaron D. Nichols, Martin Lang, Panagiotis Mitkidis.

**Writing – review & editing:** Aaron D. Nichols, Martin Lang, Christopher Kavanagh, Radek Kundt, Junko Yamada, Dan Ariely, Panagiotis Mitkidis.

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
