## [Decision Letter · Decision Letter 0]

8 Apr 2020

PONE-D-20-04631

Replicating and Extending the Effects of Auditory Religious Cues on Dishonest Behavior

PLOS ONE

Dear Mr. Nichols,

Thank you for submitting your manuscript to PLOS ONE. After careful consideration, we feel that it has merit but does not fully meet PLOS ONE’s publication criteria as it currently stands. Therefore, we invite you to submit a revised version of the manuscript that addresses the points raised during the review process.

As you will see, two reviewers have read your manuscript and they are both very positive. I agree with their assessment. This is an interesting study, the findings are straightforward and the reporting is well done! Both reviewers also make some additional suggestions to further improve the manuscript. These include mainly (1) elaborating on the distinction between religiosity / religious & ritual participation in the Introduction, (2) discussing the role of validity of single-item measures, (3) discussing the ecological validity of music priming as an experimental manipulation. As this study was not pre-registered, I think it would also be good if you could include a statement about reporting all measures that were included in the study, in this manuscript. Thanks for submitting your work to this journal and I am looking forward to receiving a revised version, addressing the points made by the reviewers. 

We would appreciate receiving your revised manuscript by May 23 2020 11:59PM. To enhance the reproducibility of your results, we recommend that if applicable you deposit your laboratory protocols in protocols.io, where a protocol can be assigned its own identifier (DOI) such that it can be cited independently in the future. For instructions see: http://journals.plos.org/plosone/s/submission-guidelines#loc-laboratory-protocols

We look forward to receiving your revised manuscript.

Kind regards,

Michiel van Elk

Academic Editor

PLOS ONE

Journal Requirements:

2) Please include captions for your Supporting Information files at the end of your manuscript, and update any in-text citations to match accordingly. Please see our Supporting Information guidelines for more information: http://journals.plos.org/plosone/s/supporting-information.

3) Please amend either the title on the online submission form (via Edit Submission) or the title in the manuscript so that they are identical.

Reviewers' comments:

Reviewer's Responses to Questions

**Comments to the Author**

1. Is the manuscript technically sound, and do the data support the conclusions?

Reviewer #1: Yes

Reviewer #2: Yes

2. Has the statistical analysis been performed appropriately and rigorously? 

Reviewer #1: Yes

Reviewer #2: Yes

3. Have the authors made all data underlying the findings in their manuscript fully available?

Reviewer #1: Yes

Reviewer #2: Yes

4. Is the manuscript presented in an intelligible fashion and written in standard English?

Reviewer #1: Yes

Reviewer #2: Yes

5. Review Comments to the Author

Reviewer #1: Summary: The authors attempt to replicate the work of Lang et al. (2016), who investigated the effects of religious instrumental music (vs. secular music or white noise) on ethical behavior (refraining from cheating). The current project is not quite a direct replication of the original work, but it is close: whereas the original research samples participants from the U.S., the Czech Republic, and Mauritius, the current work samples from the U.S., the Czech Republic, and Japan. In addition, the current work adds a no-music control condition, and conducts analyses using OLS rather than beta regression.

Lang et al. (2016) reported an interaction across all three sites between priming condition and participant religiosity, as well as a marginally significant interaction between priming condition and frequency of ritual participation. The current replication effort found no interaction between condition and religiosity, but did observe an interaction between priming condition and frequency of ritual participation, as well as an interaction between priming condition and religious affiliation.

As the field continues to focus on ensuring the robustness of findings, this is a welcome contribution. The experiments are well conducted, the manuscript is well written and well organized, and the work provides increased confidence in the claim that religious contexts and communities can influence normative behavior among participants. Overall, I recommend publication following moderate revisions, as described below.

Perhaps the most major point to be addressed concerns the theoretical connections between religiosity, frequency of ritual participation, and religious affiliation. The Introduction does not clearly delineate “religiosity” from “ritual participation”, nor does it provide a clear model of how all three of these factors are understood to relate one another. Thus, it is not entirely clear from a theoretical point of view why the authors prioritize “religiosity” as the main variable of interest, with ritual participation and religious affiliation as “supplemental” factors (the empirical justification, based on significance tests from Lang et al. [2016] is clear). Because the Introduction does not clearly disentangle the interconnections between religiosity, ritual participation, and religious affiliation, I think readers are left confused as to how they should interpret the fact that the experiments provide evidence supporting a role for the “supplemental” variables but not, apparently, the primary variable—religiosity— that seems to be the focus of the replication.

Given that the authors did not preregister the study or conduct this as a registered report, I certainly respect the way that the Introduction highlights the variable that is ultimately least supported by the data, nor should any revisions engage in HARKing. But the lack of theoretical clarity in the Introduction is a concern, and a revised introduction might elaborate upon—in a clear and more extended fashion—how religiosity is conceptually distinct from ritual participation; what the implications of this might be; and what are the possible ways that religiosity, ritual participation, and religious affiliation relate to one another and potentially interact to influence ethical behavior. This would pave the way for a revised Discussion as well, where the authors could clarify precisely what 3-way interactions they would expect to emerge in future research. Lines 481-483 of the Discussion, for instance, suggest a 3-way interaction among condition, “religiosity,” and religious affiliation. Could, or should, “frequency of ritual participation” be substituted for “religiosity” here?

A second point. The experiments are framed as a “replication.” As the recent surge of meta-science makes clear, there are many ways to define “replication” (Open Science Collaboration, 2015). Here the authors appear to adopt the approach of conducting basically the same significance tests as in the original study. This is of course completely sensible, but the manuscript would benefit from a consideration of other ways to define replication, and some treatment of the extent to which results from the current work could be considered to replicate using at least some other recommended criteria.

Third, the authors make very broad claims about “religious music” in general. At the same time, within each cultural context the studies rely upon a single example of religious music. Although the authors are hardly unusual in generalizing from a very small number of stimuli, there are serious limitations in doing so—and especially in not using statistical methods that treat stimuli as a random factor (Judd, Westfall, and Kenny, 2012). These limitations should be more prominently acknowledged in the Discussion. Doing so would also provide an opportunity to call for future research that samples multiple religious songs for each cultural context. Such a design would be well equipped to use relevant characteristics of the songs (e.g., sacredness, holiness, as well as perhaps others) to explain variation in the strength of the condition x religiosity/ritual participation interaction(s).

Some lesser points:

Abstract: The abstract comes across as too vague. Sample N’s, greater specificity concerning the origin of the participants, and a bit more detail concerning the experimental paradigm (especially control conditions) should be provided. And in discussing results, the abstract does not allude to the presence or absence of cross-cultural differences.

Introduction:

Lines 60-69. It might be worth noting that some large-scale religious priming experiments examining ethical/prosocial behavior have been conducted in the wake of van Elk’s call (e.g., White et al., 2019; Billingsley et al., 2018). Results appear to converge on the view that implicit (anagram-based) primes do not exert significant effects but more explicit (verbal/written) primes seem to exert a small effect. This might provide some useful, additional context in the Discussion as well.

Lines 153-164. Regarding sample size, it is not clear that power calculations determined the sample size. How was the sample size (460) actually determined? It’s rather disappointing that a replication study of this sort was not pre-registered, so that exclusion criteria, stopping rules, and other aspects of the protocol would have been determined ahead of time.

Line 155: Results using all participants should be made available in Supplemental.

Line 215: “They [the religious vs. secular songs] differed in perceived sacredness.” To avoid this coming across as an unjustified claim, I would alert the reader (maybe in a parenthetical insertion) that supporting data is soon to follow.

Line 218: Why does the left anchor of the religiosity scale not mention “spirituality” if the right anchor does? This could be relevant to the Discussion lines 441ff.

More attention should be paid to psychometrics and the limitations of the measures used. Most notable is the single-item measure of religiosity. Although reliability with just a single item may be a concern, I think the deeper issue is the difficulty associated with knowing that religiosity measures are functioning equivalently across cultures (Cohen et al., 2017). So, for instance, to the extent that results in Japan are driving differences between this study and the 2016 results, we can’t be sure that differential functioning of the religiosity measure might not be a contributing factor. The lack of a multi-item scale that has been tested for invariance across the sampled cultures is a limitation that should be acknowledged.

Line 295. In discussing the relatively low level of cheating observed in the Czech Republic, it might be useful to refer to Shariff and Norenzayan’s (2015) discussion of boundary conditions in the context of religious priming and prosocial behavior. The argument would be that if motivation to cheat is (for whatever reason) relatively low in the Czech Republic, the religious prime has relatively little room to operate.

Analyses: Would it make sense to include affiliation and ritual participation in the same model, to see if they predict unique variance? I think the answer depends on how a revised Introduction defines religiosity vs. ritual participation, but in principle this could be an informative analysis.

Discussion:

Lines 402-413 Seems like there should be an acknowledgment here that the “primary” hypothesis, concerning religiosity, was not supported.

Billingsley et al. (2018. Implicit and explicit influences of religious cognition on Dictator Game transfers

Royal Society open science 5 (8), 170238

Cohen et al. (2017). Theorizing and measuring religiosity across cultures. Personality and Social

Psychology Bulletin, 43(12), 1724-1736.

Judd, C. M., Westfall, J., & Kenny, D. A. (2012). Treating stimuli as a random factor in social psychology: A

new and comprehensive solution to a pervasive but largely ignored problem. Journal of

personality and social psychology, 103(1), 54.

Open Science Collaboration. (2015). Estimating the reproducibility of psychological science. Science

349(6251), aac4716.

Shariff & Norenzayan (2015). A question of reliability or of boundary conditions? Comment on Gomes

and McCullough (2015). American Psychological Association 144 (6), e105

White et al. (2019). Supernatural norm enforcement: Thinking about karma and God reduces selfishness

among believers. Journal of Experimental Social Psychology 84

Reviewer #2: I want to start by saying this is an excellent study and I think it deserves to be published basically as is. I have a few minor comments.

There is a lot of work on priming and religion and a lot that doesn’t find very robust effects, but much of this work is not looking at the nuances of what people believe and how they participant in their religion. Though I can quite confidently say that there are more than enough religious priming studies looking at sentence unscrambling tasks in Christians, there are very few looking across religions and looking at how people whose primary religious practices are based around ritual action, not belief itself. The findings here follow very clearly from the theory.

Thus, my largest criticism is that this is not stated in enough detail in the paper. The theory, and why these results should be expected, is explained in a couple sentences about how the methods differ from previous studies. This is a huge issue within the literature itself and should be addressed more fully in the paper. If we expect things like priming to work at all, it should be related to what people actually practice, rather than some method that assumes American Protestantism represent primary way people are religious. This is an important finding and should be addressed as such.

You should also include a bit more detail as to why cheating is an important measure as well. Research typically used dictator game, or generosity based measures, and the argument is that religion is enforcing norms of fairness. These norms are not consistent across cultures, but norms against cheating are. Thus, cheating is better measure for studies focusing on religious prosociality.

Data analysis is very well done. I particularly appreciate that the regression table include models with and without moderators.

Why do you only have a site by site analysis with religiosity in the supplemental? Why not add tables with ritual frequency and religious affiliation as well? These would be informative here.

Minor comments;

The Shariff et al. meta-analysis also found that religious primes are only effective on religiously affiliated people, might be worth mentioning as it supports your argument.

Relatedly, the van Elk et al. meta-analysis only failed to find an effect in the PET analysis, but did find an effect in the PEESE analysis. PET is a notoriously inaccurate meta-analytic measure (under realistic data conditions, it can fail to find a true effect between 70-90% of the time).

Table 2 is a bit messed up and hard to read in the pdf. This is probably due to rendering, but would be worth checking the original document.

6. PLOS authors have the option to publish the peer review history of their article (what does this mean?). If published, this will include your full peer review and any attached files.

Reviewer #1: No

Reviewer #2: No

---

## [Author Response · Author response to Decision Letter 0]

21 May 2020

Dear Dr. van Elk,

Thank you for inviting us to revise and resubmit our manuscript, “Replicating and Extending the Effects of Auditory Religious Cues on Dishonest Behavior” (MS PONE-D-20-04631).

We have made several changes to address the feedback provided by you and the reviewers. In particular, we have:

1) Modified our abstract to provide more details about our research methods and results

2) Clarified the key difference between religiosity and religious practice in our Introduction

3) Discussed the validity of dishonesty as a measure of religious priming effects 

4) Expanded the discussion of our research limitations, highlighting potential scale reliability issues and future opportunities to extend the literature on religious priming

5) Added table captions to our manuscript, and tables and figures to our supplement

6) Included several new references 

7) Corrected model 4 coefficients that were slightly off due to clerical errors during coding 

8) Made multiple formatting changes to align our manuscript with PLOS ONE guidelines

On the following pages, we respond in-depth to your feedback and the reviewers’ detailed comments. We think the revisions that were requested strengthen the quality of our submission and its contribution to the literature on religious priming. Please let us know if you have any remaining concerns about this work. 

Finally, in pursuit of good science and complete transparency, we have agreed to share our research materials, data, and analysis code on the Open Science Framework. These resources can be accessed at https://osf.io/k4dt8/?view_only=bb2cbb8b9c774ade984b672a3eddce43. Upon acceptance of our manuscript, these resources will be publicly available at https://osf.io/k4dt8/.

Sincerely,

Aaron D. Nichols

Martin Lang

Christopher Kavanaugh 

Radek Kundt

Junko Yamada

Panagiotis Mitkidis

Dan Ariely 

From the Editor

Dear Mr. Nichols,

Thank you for submitting your manuscript to PLOS ONE. After careful consideration, we feel that it has merit but does not fully meet PLOS ONE’s publication criteria as it currently stands. Therefore, we invite you to submit a revised version of the manuscript that addresses the points raised during the review process.

As you will see, two reviewers have read your manuscript and they are both very positive. I agree with their assessment. This is an interesting study, the findings are straightforward and the reporting is well done! Both reviewers also make some additional suggestions to further improve the manuscript. These include mainly (1) elaborating on the distinction between religiosity / religious & ritual participation in the Introduction, (2) discussing the role of validity of single-item measures, (3) discussing the ecological validity of music priming as an experimental manipulation. As this study was not pre-registered, I think it would also be good if you could include a statement about reporting all measures that were included in the study, in this manuscript. Thanks for submitting your work to this journal and I am looking forward to receiving a revised version, addressing the points made by the reviewers. 

Response: We thank the editor for inviting us to revise and resubmit our manuscript. We have revised our manuscript based on the collective feedback from the editor and the two reviewers and we believe this feedback has helped us to substantially enhance our manuscript. Specifically, we have added to the discussion of the nuances between religiosity, religious ritual participation, and affiliation in the Introduction on pages 5-6 (lines 179-229). In the Discussion section (Page 15-18, see also lines 607-612), we now detail how our research could be confounded by reliability issues. Specifically, we cite research (Cohen et al., 2017) indicating that single-item measures are susceptible to inconsistencies between tests, times, and cultures. We have also added to the Introduction (Page 2) to provide another example of auditory religious priming and we have clarified to readers why music priming can be considered an ecologically valid manipulation (Page 5-6, see also lines 194-206; lines 229-231). In our discussion (Pages 15-18, see also lines 599-607), we invite researchers to examine the generalizability of this manipulation by exploring auditory religious priming across a greater variety of culturally-relevant musical tracks. To highlight our commitment to transparency and collaborative science, we have uploaded our materials, analyses, and data to the Open Science Framework: https://osf.io/k4dt8/?view_only=bb2cbb8b9c774ade984b672a3eddce43. We also recommend the pre-registration for future studies examining these effects in our Discussion.

We would appreciate receiving your revised manuscript by May 23 2020 11:59PM. To enhance the reproducibility of your results, we recommend that if applicable you deposit your laboratory protocols in protocols.io, where a protocol can be assigned its own identifier (DOI) such that it can be cited independently in the future. For instructions see: http://journals.plos.org/plosone/s/submission-guidelines#loc-laboratory-protocols

● A rebuttal letter that responds to each point raised by the academic editor and reviewer(s). This letter should be uploaded as separate file and labeled 'Response to Reviewers'.

● A marked-up copy of your manuscript that highlights changes made to the original version. This file should be uploaded as separate file and labeled 'Revised Manuscript with Track Changes'.

● An unmarked version of your revised paper without tracked changes. This file should be uploaded as separate file and labeled 'Manuscript'.

We look forward to receiving your revised manuscript.

Kind regards,

Michiel van Elk

Academic Editor

PLOS ONE

Journal Requirements:

2) Please include captions for your Supporting Information files at the end of your manuscript, and update any in-text citations to match accordingly. Please see our Supporting Information guidelines for more information: http://journals.plos.org/plosone/s/supporting-information.

3) Please amend either the title on the online submission form (via Edit Submission) or the title in the manuscript so that they are identical.

Response: Thank you for clarifying these submission and formatting requirements. The resources you provided were very helpful to us during the revision of our manuscript. We have reformatted the manuscript to align with PLOS ONE standards, included captions for supporting information at the end of the manuscript, updated in-text citations, and have fixed the title on the online submission form to match the manuscript title. 

Reviewers' comments:

Review Comments to the Author

Reviewer #1: Summary: The authors attempt to replicate the work of Lang et al. (2016), who investigated the effects of religious instrumental music (vs. secular music or white noise) on ethical behavior (refraining from cheating). The current project is not quite a direct replication of the original work, but it is close: whereas the original research samples participants from the U.S., the Czech Republic, and Mauritius, the current work samples from the U.S., the Czech Republic, and Japan. In addition, the current work adds a no-music control condition, and conducts analyses using OLS rather than beta regression.

Lang et al. (2016) reported an interaction across all three sites between priming condition and participant religiosity, as well as a marginally significant interaction between priming condition and frequency of ritual participation. The current replication effort found no interaction between condition and religiosity, but did observe an interaction between priming condition and frequency of ritual participation, as well as an interaction between priming condition and religious affiliation.

As the field continues to focus on ensuring the robustness of findings, this is a welcome contribution. The experiments are well conducted, the manuscript is well written and well organized, and the work provides increased confidence in the claim that religious contexts and communities can influence normative behavior among participants. Overall, I recommend publication following moderate revisions, as described below.

Perhaps the most major point to be addressed concerns the theoretical connections between religiosity, frequency of ritual participation, and religious affiliation. The Introduction does not clearly delineate “religiosity” from “ritual participation”, nor does it provide a clear model of how all three of these factors are understood to relate one another. Thus, it is not entirely clear from a theoretical point of view why the authors prioritize “religiosity” as the main variable of interest, with ritual participation and religious affiliation as “supplemental” factors (the empirical justification, based on significance tests from Lang et al. [2016] is clear). Because the Introduction does not clearly disentangle the interconnections between religiosity, ritual participation, and religious affiliation, I think readers are left confused as to how they should interpret the fact that the experiments provide evidence supporting a role for the “supplemental” variables but not, apparently, the primary variable—religiosity— that seems to be the focus of the replication.

Given that the authors did not preregister the study or conduct this as a registered report, I certainly respect the way that the Introduction highlights the variable that is ultimately least supported by the data, nor should any revisions engage in HARKing. But the lack of theoretical clarity in the Introduction is a concern, and a revised introduction might elaborate upon—in a clear and more extended fashion—how religiosity is conceptually distinct from ritual participation; what the implications of this might be; and what are the possible ways that religiosity, ritual participation, and religious affiliation relate to one another and potentially interact to influence ethical behavior. This would pave the way for a revised Discussion as well, where the authors could clarify precisely what 3-way interactions they would expect to emerge in future research. Lines 481-483 of the Discussion, for instance, suggest a 3-way interaction among condition, “religiosity,” and religious affiliation. Could, or should, “frequency of ritual participation” be substituted for “religiosity” here?

Response: Thank you for identifying ways in which our theory, supplemental motivations, and introduction could be made clearer. We have expanded our discussion on the difference between religiosity and ritual participation making clear that is a nuanced issue, yet also one that is very important (Pages 5-6, lines 189-236). We have also made several changes to the Discussion (Pages 15-18), adding suggestions and comments for future research. We have also introduced some discussion of the distinction between orthodoxic and orthopraxic religious traditions which speak to the potential for important distinctions between religious beliefs and ritual practices.

A second point. The experiments are framed as a “replication.” As the recent surge of meta-science makes clear, there are many ways to define “replication” (Open Science Collaboration, 2015). Here the authors appear to adopt the approach of conducting basically the same significance tests as in the original study. This is of course completely sensible, but the manuscript would benefit from a consideration of other ways to define replication, and some treatment of the extent to which results from the current work could be considered to replicate using at least some other recommended criteria.

Response: We have added to our Introduction on page 4 (see lines 105-108) to discuss the ways in which our study is similar or unique from the various conceptualizations of replications. In our Discussion section (Pages 14-17), we also discuss how variations in the experimental context (i.e., changes to musical stimuli, culture/geographic location) could potentially influence the replicability of our results. 

Third, the authors make very broad claims about “religious music” in general. At the same time, within each cultural context the studies rely upon a single example of religious music. Although the authors are hardly unusual in generalizing from a very small number of stimuli, there are serious limitations in doing so—and especially in not using statistical methods that treat stimuli as a random factor (Judd, Westfall, and Kenny, 2012). These limitations should be more prominently acknowledged in the Discussion. Doing so would also provide an opportunity to call for future research that samples multiple religious songs for each cultural context. Such a design would be well equipped to use relevant characteristics of the songs (e.g., sacredness, holiness, as well as perhaps others) to explain variation in the strength of the condition x religiosity/ritual participation interaction(s).

Response: This is a very good point. Of course, the reviewer is right that we utilized a typical approach of priming studies by selecting stimuli that we expected would give the strongest signal at each site. Together with pre-selecting the stimuli at each site and later assessing the musical characteristics of the selected stimuli during the experiment, we aimed to amplify the tested effect. We were worried that we would not be able to isolate religious priming effects if many participants were not familiar with the stimuli. Therefore, we specifically targeted the most paradigmatic stimulus at each site. Thus, we prioritized the strength of our stimuli over generalizability, as we aimed to select the most important/known/frequently used. Of course, this does not fully ameliorate the concern of generalizability of the musical stimuli, but we hope that our pre-selection criteria and analyses, as well as our prioritization of stimuli strength allows for some degree of generalizability. To make these points clearer to the reader, we now discuss the issue of generalizability (on Pages 17; see also lines 602-606) and suggest that future researchers should investigate religious priming effects across a more diverse array of culturally relevant musical tracks (e.g., by selecting the three most salient musical stimuli at each site and randomly vary those stimuli across participants). We have also added the reference that you provided to make our stimuli limitations clearer. 

Some lesser points:

Abstract: The abstract comes across as too vague. Sample N’s, greater specificity concerning the origin of the participants, and a bit more detail concerning the experimental paradigm (especially control conditions) should be provided. And in discussing results, the abstract does not allude to the presence or absence of cross-cultural differences.

Response: We appreciate the reviewer’s suggestions for improving our abstract. We have revised our abstract (Page 2) to give the readers more details about the design and findings of this research. We hope the key details that were added enhance our abstract’s clarity and quality. 

Introduction:

Lines 60-69. It might be worth noting that some large-scale religious priming experiments examining ethical/prosocial behavior have been conducted in the wake of van Elk’s call (e.g., White et al., 2019; Billingsley et al., 2018). Results appear to converge on the view that implicit (anagram-based) primes do not exert significant effects but more explicit (verbal/written) primes seem to exert a small effect. This might provide some useful, additional context in the Discussion as well.

Response: Thank you for bringing our attention to these key references. Indeed, these references were quite helpful in adding context to our Introduction (Pages 2-6) and to our Discussion (Page 15-18). These references helped us further clarify how our research relates to recent contributions in the literature on religious priming.

Lines 153-164. Regarding sample size, it is not clear that power calculations determined the sample size. How was the sample size (460) actually determined? It’s rather disappointing that a replication study of this sort was not pre-registered, so that exclusion criteria, stopping rules, and other aspects of the protocol would have been determined ahead of time.

Response: We appreciate the opportunity to discuss our sample in greater detail. We used G*Power to calculate the estimated power of our sample and the R^2 increase from Model1 to Model2 as reported in Table 2 in Lang et al., 2016. Here is the screenshot of the G*Power setup:

Of course, this approach is a bit imprecise because in the current study, our model includes one more experimental condition. On the other hand, this additional condition should yield higher variance explained by the added predictors and counterbalance the increased number of predictors. Indeed, re-running the same power calculation with the assumed additional predictor, we need the variance explained by special effect to be raised only to 0.025 to have power of 0.85 for our sample size.

Furthermore, we share the reviewer’s disappointment that the study was not pre-registered. Regrettably, we did not utilize pre-registration methods when this study was originally incepted. In hindsight, this would be an obvious thing to do now, but we cannot go back in time. However, in pursuit of transparency and healthy science, we have made our resources, data, and analysis code available. For the sake of transparency, we have added a data availability statement (Page 19) indicating that our measures, analyses, and data have been made publicly available at OSF (https://osf.io/k4dt8/?view_only=bb2cbb8b9c774ade984b672a3eddce43.).

Line 155: Results using all participants should be made available in Supplemental.

Response: Thank you for this suggestion. We now detail results using all participants in a new supplemental file. In our Methods section (Page 7, lines 244-248), we direct readers to view our results on all participants in a supplement (See Table G in S1). We have also made all of our results and materials available at OSF (https://osf.io/k4dt8/?view_only=bb2cbb8b9c774ade984b672a3eddce43).

Line 215: “They [the religious vs. secular songs] differed in perceived sacredness.” To avoid this coming across as an unjustified claim, I would alert the reader (maybe in a parenthetical insertion) that supporting data is soon to follow.

Response: Thank you for sharing this advice. We have now amended this wording (Line 307) to appropriately convey our results to readers. 

Line 218: Why does the left anchor of the religiosity scale not mention “spirituality” if the right anchor does? This could be relevant to the Discussion lines 441ff.

Response: Thank you for alerting us to this scale asymmetry. Indeed, the left anchor of the religiosity scale does not mention spirituality. This asymmetry was likely a mistake that occurred when we were adapting the materials of Lang et. al, 2016. Unfortunately, we missed this clerical error and this version of the scale was implemented in all three sites. In light of this, we now discuss this scale limitation in the Discussion section (Pages 15-18) how the inclusion of spirituality at only the right anchor may have impacted our ability to replicate the religiosity*condition interaction observed in Lang et. al, 2016. Further, we encourage future researchers to extend and improve on our research by explicitly testing the reliability of religiosity scales prior to conducting religious priming research. 

More attention should be paid to psychometrics and the limitations of the measures used. Most notable is the single-item measure of religiosity. Although reliability with just a single item may be a concern, I think the deeper issue is the difficulty associated with knowing that religiosity measures are functioning equivalently across cultures (Cohen et al., 2017). So, for instance, to the extent that results in Japan are driving differences between this study and the 2016 results, we can’t be sure that differential functioning of the religiosity measure might not be a contributing factor. The lack of a multi-item scale that has been tested for invariance across the sampled cultures is a limitation that should be acknowledged.

Response: We appreciate this important point about item-reliability (especially across cultures and languages). In our Discussion section (page 15-18, see also lines 607-619), we now highlight the limitation of single-item scales and how single-item measures are susceptible to reliability issues. In light of these limitations, we now invite future researchers (Page 17) to replicate our results using a scale that has been explicitly tested for reliability across time, tests, and samples. We have also included the references provided to provide the reader more information on issues with cross-cultural item reliability. 

Line 295. In discussing the relatively low level of cheating observed in the Czech Republic, it might be useful to refer to Shariff and Norenzayan’s (2015) discussion of boundary conditions in the context of religious priming and prosocial behavior. The argument would be that if motivation to cheat is (for whatever reason) relatively low in the Czech Republic, the religious prime has relatively little room to operate.

Response: Thank you for sharing this critical insight. On Pages 17 (see also lines 594-599), we now discuss how boundary conditions, citing Shariff and Norenzayan’s commentary (2015), may partly explain the relatively low level of cheating observed in the Czech Republic sample.

Analyses: Would it make sense to include affiliation and ritual participation in the same model, to see if they predict unique variance? I think the answer depends on how a revised Introduction defines religiosity vs. ritual participation, but in principle this could be an informative analysis.

Response: We appreciate the reviewer’s thought-provoking suggestion. Ultimately, we have opted to minimize model noise and to keep affiliation and ritual participants in separate models. We agree that religiosity, religious affiliation, and ritual participation are distinct concepts (see revised Introduction; especially lines 189-236). However, despite the important distinction between affiliation and ritual participation, we also note that they are highly correlated in our sample and that their joint inclusion in a model would cause issues with multicollinearity. Hence, we treat them separately but encourage other researchers to measure and investigate the relationships we report in independent samples.

Discussion:

Lines 402-413 Seems like there should be an acknowledgment here that the “primary” hypothesis, concerning religiosity, was not supported.

Response: Thank you, this is an important suggestion. We have added a line to our Discussion (508-509) to explicitly mention that the primary hypothesis was not supported. 

Billingsley et al. (2018. Implicit and explicit influences of religious cognition on Dictator Game transfers. Royal Society open science 5 (8), 170238

Cohen et al. (2017). Theorizing and measuring religiosity across cultures. Personality and Social Psychology Bulletin, 43(12), 1724-1736.

Judd, C. M., Westfall, J., & Kenny, D. A. (2012). Treating stimuli as a random factor in social psychology: A new and comprehensive solution to a pervasive but largely ignored problem. Journal of personality and social psychology, 103(1), 54.

Open Science Collaboration. (2015). Estimating the reproducibility of psychological science. Science 349(6251), aac4716.

Shariff & Norenzayan (2015). A question of reliability or of boundary conditions? Comment on Gomes and McCullough (2015). American Psychological Association 144 (6), e105

White et al. (2019). Supernatural norm enforcement: Thinking about karma and God reduces selfishness among believers. Journal of Experimental Social Psychology 84

Response: Thank you for providing these key resources and citations. We believe their inclusion, as well as addressing the points you raised above, have strengthened our paper. 

Reviewer #2: I want to start by saying this is an excellent study and I think it deserves to be published basically as is. I have a few minor comments.

There is a lot of work on priming and religion and a lot that doesn’t find very robust effects, but much of this work is not looking at the nuances of what people believe and how they participant in their religion. Though I can quite confidently say that there are more than enough religious priming studies looking at sentence unscrambling tasks in Christians, there are very few looking across religions and looking at how people whose primary religious practices are based around ritual action, not belief itself. The findings here follow very clearly from the theory.

Thus, my largest criticism is that this is not stated in enough detail in the paper. The theory, and why these results should be expected, is explained in a couple sentences about how the methods differ from previous studies. This is a huge issue within the literature itself and should be addressed more fully in the paper. If we expect things like priming to work at all, it should be related to what people actually practice, rather than some method that assumes American Protestantism represent primary way people are religious. This is an important finding and should be addressed as such.

Response: Thank you for raising these important points. We have added to our Introduction (Pages 2-6; see also 189-236) to address these points directly. Specifically, we develop our theory, predictions, and establish the importance of our research by discussing the limited scope of religious priming literature. We have added key references, and discuss how the research literature could benefit from investigations that delineate under-explored, yet important, aspects of religious practice such as ritual participation, affiliation to a specific religious organization, and other unmeasured factors that vary across cultures (labeled as the WEIRD people problem). 

You should also include a bit more detail as to why cheating is an important measure as well. Research typically used dictator game, or generosity based measures, and the argument is that religion is enforcing norms of fairness. These norms are not consistent across cultures, but norms against cheating are. Thus, cheating is better measure for studies focusing on religious prosociality.

Response: Thank you for sharing this interesting perspective and for encouraging us to explain our use of cheating measures. We agree with your points and have added to our Introduction (Page 2-6; see Lines 89-92, 128-132) and to our Discussion (page 14-17; see lines 624-626) make these clear to the reader. As part of this added discussion, we include references highlighting the inconsistency of fairness norms (measured by dictator games) across cultures. 

Data analysis is very well done. I particularly appreciate that the regression table include models with and without moderators.

Why do you only have a site by site analysis with religiosity in the supplemental? Why not add tables with ritual frequency and religious affiliation as well? These would be informative here.

Response: Thank you for this suggestion. We opted to replicate tables presented in Lang et al. 2016 so we included the site by site analysis in the supplemental material. To better inform the readers, we now include tables with ritual frequency and religious affiliation in our supplemental material (See Tables B and C in S1). 

Minor comments;

The Shariff et al. meta-analysis also found that religious primes are only effective on religiously affiliated people, might be worth mentioning as it supports your argument.

Relatedly, the van Elk et al. meta-analysis only failed to find an effect in the PET analysis, but did find an effect in the PEESE analysis. PET is a notoriously inaccurate meta-analytic measure (under realistic data conditions, it can fail to find a true effect between 70-90% of the time).

Response: Thank you very much for bringing up these points. We believe the point about the Shariff et al. meta-analysis enhances the context for our supplemental theory and results. We have added a discussion of this point our Introduction (2-6; see lines 75 and 80-87). We have also added the point about PET analyses, with supporting citations (Stanley, 2017; Carter, Schonbrodt, Gervais, & Hilgard, 2019), in our Introduction (Pages 2-6). We hope this point and the van Elk et al. (2015) findings further illustrates the importance of replications. 

Table 2 is a bit messed up and hard to read in the pdf. This is probably due to rendering, but would be worth checking the original document.

Response: Thank you for alerting us to this issue. Unfortunately, we believe this may be a rendering issue. However, this table can now also be accessed on OSF, potentially fixing this problem. (OSF link: https://osf.io/k4dt8/?view_only=bb2cbb8b9c774ade984b672a3eddce43)

---

## [Decision Letter · Decision Letter 1]

20 Jul 2020

Replicating and Extending the Effects of Auditory Religious Cues on Dishonest Behavior

PONE-D-20-04631R1

Dear Dr. Nichols,

We’re pleased to inform you that your manuscript has been judged scientifically suitable for publication and will be formally accepted for publication once it meets all outstanding technical requirements.

Kind regards,

Michiel van Elk

Academic Editor

PLOS ONE

Additional Editor Comments (optional):

Reviewers' comments:

Reviewer's Responses to Questions

**Comments to the Author**

1. If the authors have adequately addressed your comments raised in a previous round of review and you feel that this manuscript is now acceptable for publication, you may indicate that here to bypass the “Comments to the Author” section, enter your conflict of interest statement in the “Confidential to Editor” section, and submit your "Accept" recommendation.

Reviewer #1: All comments have been addressed

Reviewer #2: All comments have been addressed

2. Is the manuscript technically sound, and do the data support the conclusions?

Reviewer #1: (No Response)

Reviewer #2: Yes

3. Has the statistical analysis been performed appropriately and rigorously? 

Reviewer #1: (No Response)

Reviewer #2: Yes

4. Have the authors made all data underlying the findings in their manuscript fully available?

Reviewer #1: (No Response)

Reviewer #2: Yes

5. Is the manuscript presented in an intelligible fashion and written in standard English?

Reviewer #1: (No Response)

Reviewer #2: Yes

6. Review Comments to the Author

Reviewer #1: In my view, the authors have successfully addressed the comments raised by reviewers, and the result is a strong contribution to our understanding of religious practice in relation to ethical behavior across cultures. Of particular quality were the revisions to the Introduction and Discussion, which considerably clarified the inter-relationships among religiosity, religious attendance, and religious affiliation, and how those factors differentially pertain to the effect of religious cues—especially musical ones—on prosocial behavjor. The additions pertaining to the study’s limitations and directions for future research, as well as more thorough contextualizing of the study with respect to other recent large-scale priming studies, were also well executed. Altogether, the authors have been thorough and conscientious in revising the manuscript in light of prior comments. I certainly feel that the resulting paper is considerably improved. I hope the authors do as well, and I recommend publication.

Reviewer #2: The authors have addressed all my concerns, and I recommend acceptance of this paper. I very much enjoyed reading this.

7. PLOS authors have the option to publish the peer review history of their article (what does this mean?). If published, this will include your full peer review and any attached files.

Reviewer #1: No

Reviewer #2: No

---

## [Editor Report · Acceptance letter]

30 Jul 2020

PONE-D-20-04631R1 

Replicating and Extending the Effects of Auditory Religious Cues on Dishonest Behavior 

Dear Dr. Nichols:

I'm pleased to inform you that your manuscript has been deemed suitable for publication in PLOS ONE. Congratulations! Your manuscript is now with our production department. 

Kind regards, 

on behalf of

Dr. Michiel van Elk 

Academic Editor

PLOS ONE